# Time of Flight Size Control of Carbon Nanoparticles Using Ar+CH₄ Multi-Hollow Discharge Plasma Chemical Vapor Deposition Method

Sung Hwa Hwang [1], Kazunori Koga [1,2,*], Yuan Hao [1], Pankaj Attri [3], Takamasa Okumura [1], Kunihiro Kamataki [1], Naho Itagaki [1], Masaharu Shiratani [1], Jun-Seok Oh [4], Susumu Takabayashi [5] and Tatsuyuki Nakatani [6]

1 Department of Electronics, Kyushu University, Fukuoka 819-0395, Japan; sh.hwang@plasma.ed.kyushu-u.ac.jp (S.H.H.); y.hao@plasma.ed.kyushu-u.ac.jp (Y.H.); t.okumura@plasma.ed.kyushu-u.ac.jp (T.O.); kamataki@plasma.ed.kyushu-u.ac.jp (K.K.); itagaki@ed.kyushu-u.ac.jp (N.I.); siratani@ed.kyushu-u.ac.jp (M.S.)
2 Center for Novel Science Initiatives, National Institutes of Natural Science, Tokyo 105-0001, Japan
3 Center of Plasma Nano-Interface Engineering, Kyushu University, Fukuoka 819-0395, Japan; attri.pankaj.486@m.kyushu-u.ac.jp
4 Graduate School of Engineering, Osaka City University, Osaka 558-8585, Japan; jsoh@osaka-cu.ac.jp
5 National Institute of Technology, Ariake College, Fukuoka 836-8585, Japan; stak@ariake-nct.ac.jp
6 Institute of Frontier Science and Technology, Okayama University of Science, Okayama 700-0005, Japan; nakatani@bme.ous.ac.jp
* Correspondence: koga@ed.kyushu-u.ac.jp

**Abstract:** As the application of nanotechnology increases continuously, the need for controlled size nanoparticles also increases. Therefore, in this work, we discussed the growth mechanism of carbon nanoparticles generated in Ar+CH₄ multi-hollow discharge plasmas. Using the plasmas, we succeeded in continuous generation of hydrogenated amorphous carbon nanoparticles with controlled size (25–220 nm) by the gas flow. Among the nanoparticle growth processes in plasmas, we confirmed the deposition of carbon-related radicals was the dominant process for the method. The size of nanoparticles was proportional to the gas residence time in holes of the discharge electrode. The radical deposition developed the nucleated nanoparticles during their transport in discharges, and the time of flight in discharges controlled the size of nanoparticles.

**Keywords:** plasma chemical vapor deposition; carbon nanoparticle; coagulation; optical emission spectroscopy





## 1. Introduction

Carbon nanoparticles (CNPs) have attracted tremendous attention for their various applications, such as electrical conductivity improvement of polymer, lubrication applications, cancer cell treatments, bioimaging diagnostics [1–3]. Therefore, it is essential to develop a simple method to control the size and structure of CNPs [4–7]. The solution process is a conventional method of producing CNPs, but this method has limitations like impurity, unexpected agglomeration, and low throughput due to the multistage process [8–11].

The plasma process plays a promising role because it is a dry process using low pressure resulting from reducing impurity and avoid agglomeration or coagulation due to the charge of CNPs. However, the traditional plasma process has a problem regarding throughput due to pulsed discharges for size control [12–14]. Traditional plasma process has the discharge off period to wait for pumping out the particles from the gas phase, resulting in lower throughput.

To date, we have successfully synthesized Si NPs and CNPs by using multi-hollow discharge plasma chemical vapor deposition (MHDPCVD), which can be produced continuously by employing fast gas flow [15–23]. In this method, the gas flow direction is

uniform in the plasma region, and NPs are nucleated and grown in plasma. The nucleated NPs were transported toward the outside of the plasmas by viscous gas force. That results in stopped growth outside of the plasma, which helped in the continuous production of size-controlled NPs.

NPs synthesis by the MHDPCVD method undergoes parametric tests such as dependence on gas pressure, gas flow rate, and gas composition, which are the external parameters [15–24]. Using the MHDPCVD, crystalline Si nanoparticles of 2 nm in size with 0.5 nm in size dispersion were produced for nanocrystalline amorphous silicon films for the third generation solar cells [15–21]. We employed two MHDPCVD sources to produce size-controlled Si nanoparticles and to cover nitrogen on the particles. The surface-modified nanoparticles showed multi-exciton generation, which is necessary to increase solar cells' efficiency [18]. We recently used MHDPCVD to produce carbon nanoparticles [23–25] and confirmed that pressure played an important role in size control [23]. These studies revealed the essential parameters for the size control, while the growth mechanism was unclear. Hence, in this study, we measured the CNPs size dependence on the gas flow rate (*FR*) and discussed the growth mechanism of nanoparticles produced by the MHDPCVD method.

## 2. Materials and Methods

Figure 1 illustrates a schematic diagram of the MHDPCVD reactor [23,24]. Powered and grounded electrodes have 8 holes of 5 mm diameter. The powered electrode of 5 mm in thickness was a sandwich between two grounded electrodes of 1 mm in thickness. The gap between the powered and grounded electrode was 2 mm, and the total length of a hole was 11 mm. Ar and $CH_4$ gases were introduced from the chamber's left side, passed through the holes, and later evacuated by the pump system. The *FR* ratio of Ar and $CH_4$ was 6:1. The total *FR* was controlled in a range of 10–120 sccm. During this process, gas pressure was kept at 266 Pa. The substrate holder was set at 100 mm apart from the electrode in the downstream region, and it was grounded. The powered electrode was connected to a 60 MHz radio frequency (rf) power supply through an impedance matching box. The discharge power and discharge period were 40 W and 90 min, respectively, and corresponding discharge and self-bias voltages were 230 and 80 V, respectively.

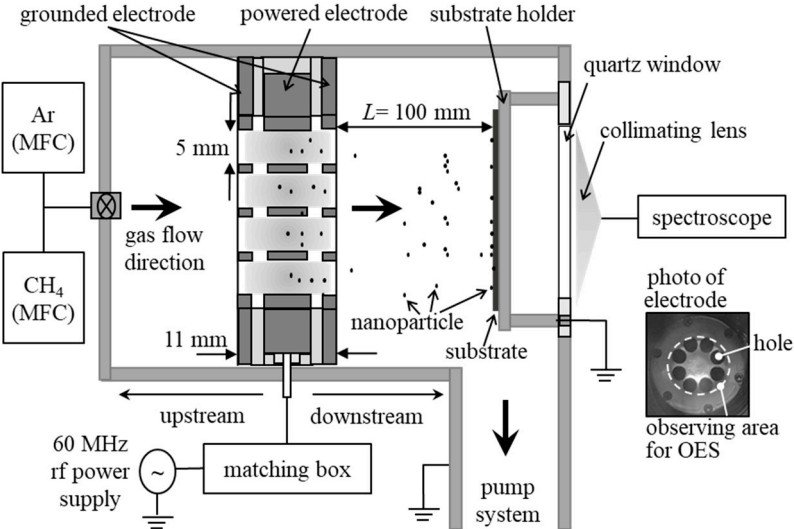

**Figure 1.** Multi-hollow discharge plasma.

The CNPs generated in the discharges were collected by using mesh grids for a transmission electron microscope (TEM) and Si substrates for a Raman spectroscopy. The size and structure of CNPs were measured with TEM (JEOL, JEM-2010) and Raman spectroscope (Jasco, NRS3000; λ = 532 nm), respectively. Optical emission from plasmas

discharges (all eight holes) was monitored by spectroscope (Ocean Optics, USB2000+) equipped with a collimating lens.

## 3. Results and Discussion

Figure 2a–d show the TEM images of CNPs as an FR parameter. With increasing *FR*, the mean size of CNPs decreased from approximately 220 nm at *FR* = 10 sccm to 25 nm at *FR* = 120 sccm. For *FR* = 10 sccm (Figure 2a) and *FR* = 20 sccm (Figure 2b), the CNPs deposit sparsely, while they deposit densely for *FR* = 50 sccm (Figure 2c) and *FR* = 120 sccm (Figure 2d). Additionally, the number of deposited CNPs increased with increasing *FR*. This shows the flux of CNPs increases with the increase in *FR*. Above 50 sccm, the deposited CNPs were stacked, then the absolute number of the deposited CNPs were unclear. Thus, we have evaluated the probability distribution of the CNPs for each *FR*.

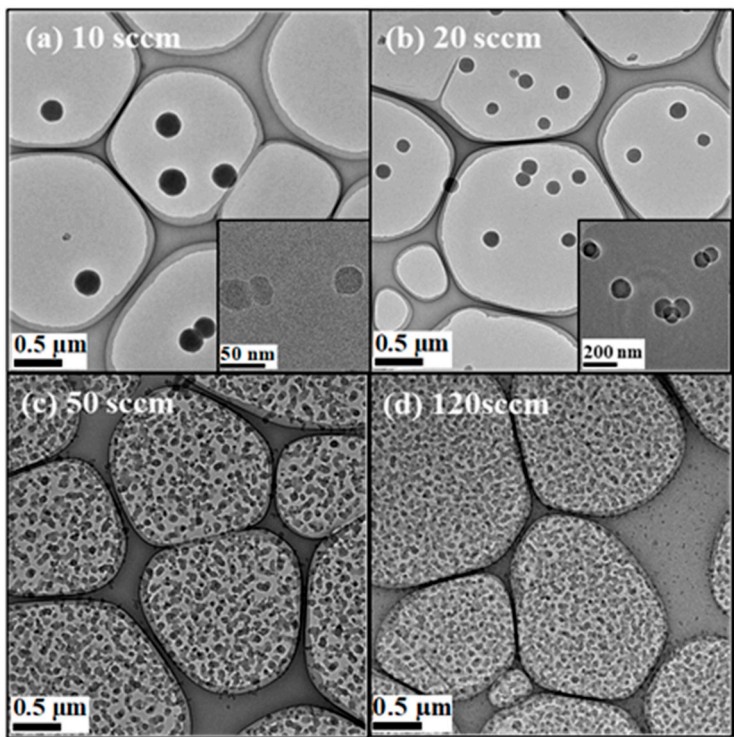

**Figure 2.** TEM images of carbon nanoparticles produced for (**a**) *FR* = 10 sccm, (**b**) *FR* = 20 sccm, (**c**) *FR* = 50 sccm, and (**d**) *FR* = 120 sccm. Insets in (**a**) and (**b**) show their high magnification TEM images.

Figure 3 shows the size distribution of the deposited CNPs obtained from TEM images where $d_p$ is the CNP size (diameter). The deposited nanoparticles were stacked for *FR* above 50 sccm. Thus, we estimated the probability of CNPs deposited on the mesh grid. Two group sizes were produced for *FR* = 10 sccm; (1) smaller group size has a size range between 20 and 90 nm, and (2) larger group size has a range between 170 and 250 nm. For *FR* = 20 sccm, two peaks at 60 and 150 nm were detected, but these peaks overlap and form one size group with a wide range between 30–200 nm. At the same time, one group size was obtained for *FR* above 50 sccm. Therefore, as the *FR* increases from 50 to 120 sccm, the peak size gradually shifts toward a smaller size from 45 to 20 nm, respectively. The size dispersion became narrower for higher *FR* from 50 and 120 sccm.

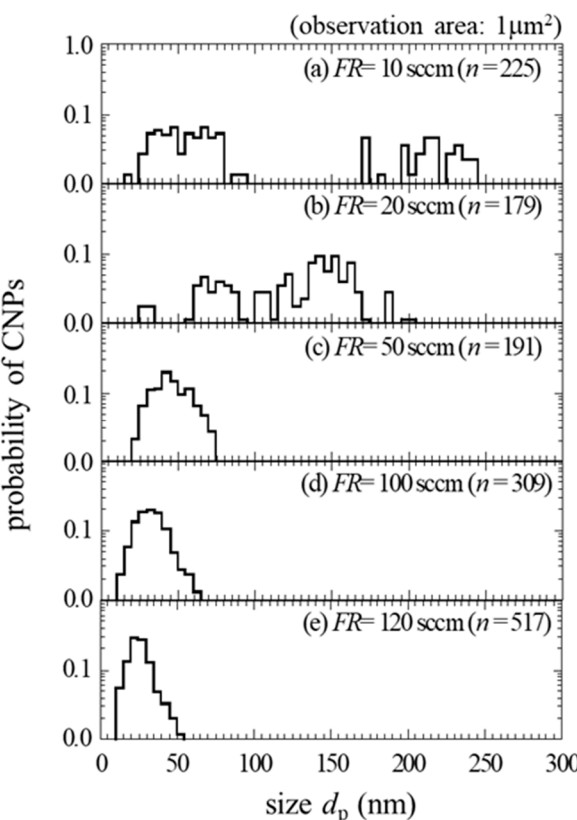

**Figure 3.** The size distribution of CNPs as a parameter of *FR* where n is the number of measured CNPs.

From the size distribution in Figure 3, we plotted a dependence of $d_p$ on *FR*, as shown in Figure 4. At *FR* below 20 sccm, the larger-sized nanoparticles seem to be separated from the smaller size group and grow in a monodisperse way. Similar growth behavior was observed for Si nanoparticles in silane plasmas in the earlier study [26]. Considering the larger size of CNPs at *FR* = 10 sccm, the $d_p$ decreases monotonically with increasing *FR*.

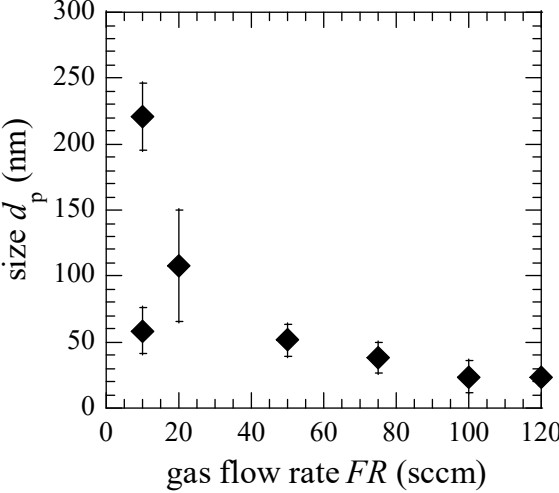

**Figure 4.** Dependence of $d_p$ on *FR*. Error bar shows the standard deviation.

To obtain the structure of the CNPs, we have measured the XRD and Raman spectra of CNPs deposited at *FR* = 50 sccm. Figures 5 and 6 show the XRD and Raman spectra, respectively. A broad peak in the XRD spectrum appears around $2\theta = 20°$ and corresponds to the hydrogenated amorphous carbon (a-C:H) [27]. Figure 6 shows a Raman spectrum of nanoparticles

deposited on the Si substrates at 50 sccm *FR*. Raman spectra clearly show the separated D (1350 cm$^{-1}$), and G (1580 cm$^{-1}$) bands. The area intensity ratio of D/G band was around 1.8; this indicates the structure of the CNPs were polymer like a-C:H [25,28–30]. Similar spectra were also observed at other *FR*s.

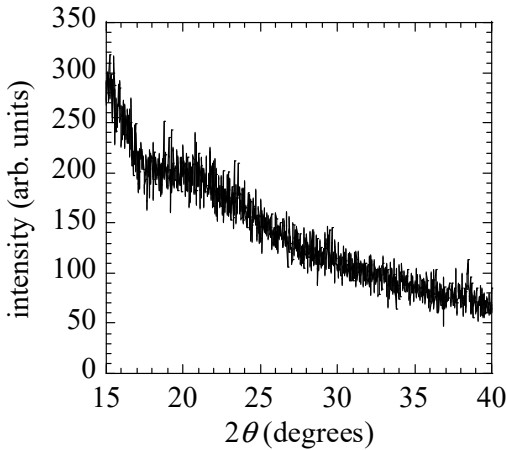

**Figure 5.** XRD spectrum of CNPs for *FR* = 50 sccm.

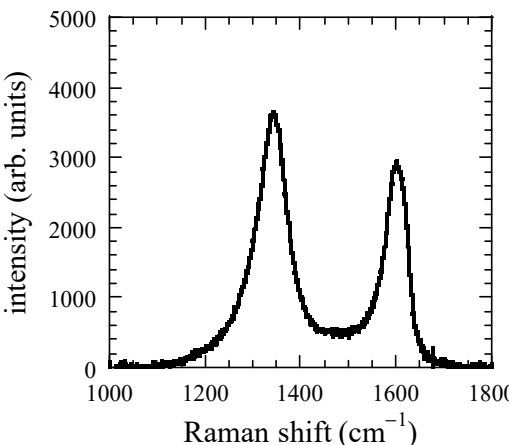

**Figure 6.** Raman spectrum of CNPs for *FR* = 50 sccm.

For the nanoparticle growth in the conventional CCP, the discharge duration is an essential factor. The size of nanoparticles increasing with an increase in CCP discharge duration [31]. The discharge duration was related to the period, which is the sum of the nucleation time and subsequent nucleated nanoparticles' growth time. Continuous discharges sustained in holes and a low-density plasma penetrated the holes due to high working pressure of 266 Pa. The generated CNPs were transported inside the holes by the gas flow. The growth time of CNPs in the plasmas correlates with the gas residence time in holes. The gas residence time $\tau_{\text{res}}$ of holes corresponds to discharge duration in the conventional CCP. In this study, gas residence time was calculated from *FR*. For the CNP, growth involves two growth processes like the coagulation of CNPs during transport toward substrates and radical deposition on CNPs.

For the coagulation, CNPs are grown by the collision between two CNPs, as the volume of CNPs after the collision is the sum of two CNPs volume (before the collision). The size $d_{\text{p1}}$ and number density $n_{\text{p1}}$ of CNPs after the collision are expressed by $d_{\text{p1}} = 2^{\frac{1}{3}} d_{\text{p0}}$ and $n_{\text{p1}} = n_{\text{p0}} - 1$, respectively, where $d_{\text{p0}}$ and $n_{\text{p0}}$ are the size and density of CNPs before the collision. To figure out the effects of the coagulation of CNPs, we examined the CNPs deposition at three positions in the transport region. Figure 7 shows the dependence of

the size and surface density of deposited CNPs on the position *L* far from the electrode. For *L* = 100, 120, and 140 mm, the size is irrelevant to the position. The area density monotonically decreases with increasing *L*.

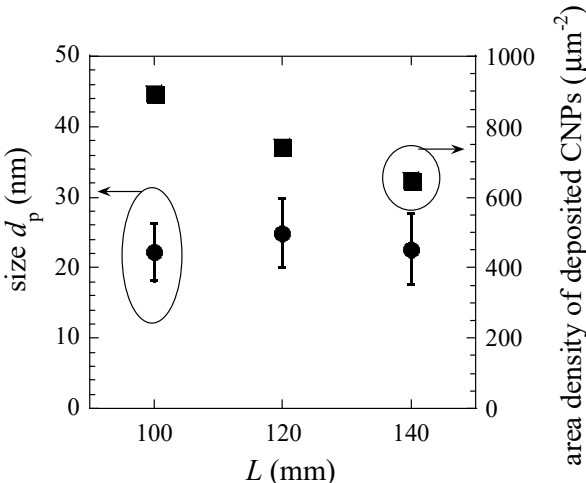

**Figure 7.** Dependence of the size and surface density of deposited CNPs on the position *L* far from the electrode for *FR* = 100 sccm. Error bar shows the standard deviation.

Previously, it was reported that the flux of CNPs proportionally increases with increasing the solid angle in the multi-hollow discharge plasma CVD method [24]. The solid angle is the main factor of decreased area density, see Figure 7. The results of the size and the area density indicate that the coagulation of CNPs was negligible. For the radical deposition, the growth rate $G_r$ of CNP expressed as equation 1

$$G_r = \frac{dd_p}{dt} = 2DR_r, \tag{1}$$

where $d_p$ is the size (diameter) of CNPs and $DR_r$ is the deposition rate of radicals on CNPs. If we assume the sticking probability of radicals on CNPs is unity and carbon atoms are responsible for the mass of CNPs, the $G_r$ is given by

$$G_r = \frac{dd_p}{dt} = \frac{2}{\rho} m_C n_r v_{thr}, \tag{2}$$

where $\rho$ is the mass density of CNPs, $m_C$ the mass of a carbon atom ($2.00 \times 10^{-26}$ kg), $n_r$ the number density of the radicals in plasmas, and $v_{thr}$ the thermal velocity of the radicals. The size and density of CNPs affect the radical density. The loss of radicals to the chamber wall is dominant if their size and density are low, while the loss to CNPs is prevalent if their size and density are high. The loss mode is determined by the coupling parameter $\Gamma$ of CNPs in plasmas [32], given by the following equation.

$$\Gamma = \frac{1}{6} d_p^2 n_p^{\frac{5}{3}} D_w^3, \tag{3}$$

where $D_w$ is the characteristic length of the reactor. For the $\Gamma \gg 1$, the coupling among CNPs through radicals is strong, results in the deceased radical density with the time after the nucleation of CNPs. If the coupling is weak, the wall loss of radicals is dominant, resulting in no radical density change with the time. Further, to detect the $\Gamma$ value, the $n_p$ was deduced from the result in Figure 7.

Figure 7 shows the number of deposited CNPs per μm$^2$ during the deposition time of 60 min. The flux of the CNPs can be calculated if the sticking probability of CNPs is unity. Considering the solid angle, the flux at the end of the holes deduced to be

$1.30 \times 10^{11}$ cm$^{-2}$s$^{-1}$. Raman results show that the structure of CNPs was polymer-like carbon, and the mass density of the CNPs was assumed 1.6 g/cm$^3$. If the temperature of CNPs equal to that of the electrode (433 K), the $n_p$ was $1.20 \times 10^9$ cm$^{-3}$, and $d_p$ was 25 nm, as shown in Figure 6. $D_w$ ($D_w$ = 2.5 mm) assumed as the radius of hole, then $\Gamma$ was 6.78 at the end of the discharge region. In the discharge region, the size of CNPs was smaller than 25 nm, and $\Gamma$ value should be less than one. It suggests that the loss of radicals through the wall was predominant, which results in a constant rate of radical loss. To discuss the generation of the radicals, we have measured emission spectra in plasma. We measured two Ar I emission intensities at 425.9 nm $I_{425.9}$ and 750.4 nm $I_{750.4}$ with upper-level excitation energy of 14.7 eV ($3p_1$) and 13.5 eV ($2p_1$), respectively. These emission processes have little effect on quenching and radiation trapping. The upper excitation level has small cross sections for electron-impact excitation from metastable states [33,34]. The *FR* dependence of an emission intensity ratio $I_{425.9}/I_{750.4}$, shown in Figure 8. The ratio indicates the information of the high energy tail of the electron energy distribution, which relates to the radical generation.

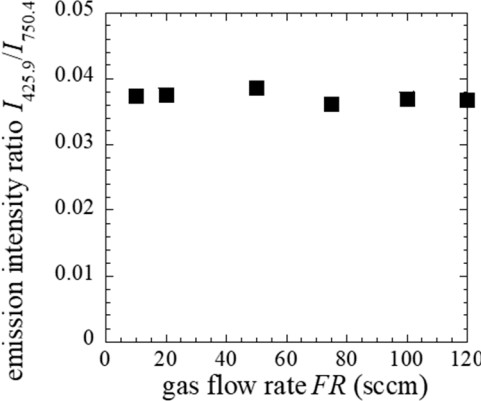

**Figure 8.** Dependence of $I_{425.9}/I_{750.4}$ on *FR*.

Although the ratio is irrelevant to *FR*, as shown in Figure 8, the discharge voltage for each *FR* condition is almost the same, suggesting the electron density was irrelevant to the *FR*. These results indicate that the generation rate of radicals is unrelated to the *FR*.

In the steady state, the $n_r$ is proportional to the density of CH$_4$ because the electron density and the loss rate of the radicals can be assumed to be constant based on the above discussion. Integrating Equation (2), the following formula gives the CNP size.

$$d_p = \frac{2}{\rho} k m_C n_{CH4} v_{thr} t, \tag{4}$$

where $k$ is the ratio of generation rate and loss rate of radicals in the steady state, $n_{CH4}$ the density of CH$_4$, and $t$ the interaction time of CNPs and radicals. The $k$ value is related to the depletion rate of the CH$_4$ molecules. In the current study, CNPs were nucleated in the discharge generated in the holes of the electrode. They grew in the discharge, transport with the gas flow, and growth was stopped outside the holes. We assumed that the growth of CNPs starts when the CH$_4$ molecules enter into the holes where plasmas were generated. Thus, $d_p$ was assumed to be equal to zero at $t = 0$, and the growth of CNPs stops at $t = \tau_{res}$. Figure 8 shows the dependence of $d_p$ on $\tau_{res}$, based on *FR* dependence, together with the results reported earlier [23]. Considering the larger size of CNPs for *FR* = 10 sccm, the size of CNPs linearly increases with increasing the $\tau_{res}$. In this study, $n_{CH4}$ and $v_{thr}$ was $6.36 \times 10^{21}$ m$^{-3}$ and $8.24 \times 10^2$ m/s, respectively. The calculated value using Equation (4) as a parameter of $k$ and the experimental results were well fitted for $k = 0.035$ (Figure 9). For the conventional CCP, the depletion rate of CH$_4$ was about 3% for

1.33 Pa pure $CH_4$ gas and 0.15 W/cm$^2$ in discharge power density [35]. The depletion rate monotonically increases with $CH_4$ pressure.

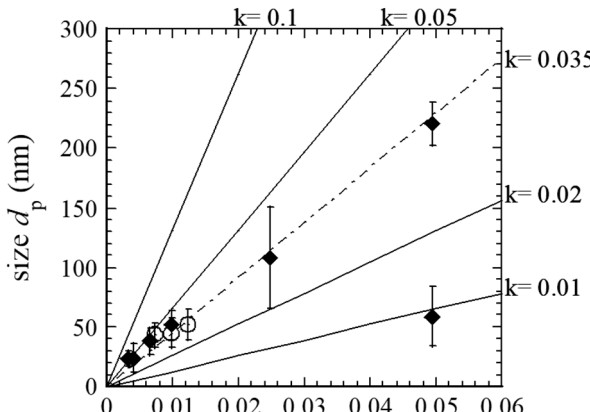

**Figure 9.** Dependence of $d_p$ on $\tau_{res}$. Open circles indicate the results reported earlier [23]. Lines were obtained by Equation (4). Error bar shows the standard deviation.

For the MHDPCVD, the discharge power density was 6.4 W/cm$^2$, much higher than the conventional plasma CVD (above-mentioned), and the partial pressure of $CH_4$ was 38 Pa. The radical loss to CNPs was small but cannot be ignored as it affects the $\Gamma$ value. Thus, the fitted value of $k$ is reasonable. Based on our results, the CNP in MHDPCVD was grown by the deposition of carbon-related radicals.

## 4. Conclusions

Through this work, we succeeded in synthesizing the size controlled CNPs using the Ar +CH4 MHDPCVD method continuously. The control range of the mean size was from 25 to 220 nm. We observed that size was proportional to the gas residence time in the discharges maintained in the electrode's holes. We theoretically confirmed that CNPs were grown by the deposition of radicals during the discharges' transport of CNPs, and CNPs move through gas flow in the discharges. The duration of the CNP transport in the discharge corresponds to the gas residence time. Therefore, the CNP size control using the MHDPVCD is a type of time of flight size control.

**Author Contributions:** Formal analysis, S.H.H. and Y.H.; writing—original draft preparation, S.H.H. and K.K. (Kazunori Koga); writing—review and editing, P.A., T.O., K.K. (Kunihiro Kamataki), N.I., M.S., J.-S.O., S.T. and T.N.; supervision, K.K. (Kazunori Koga), M.S. and T.N.; funding acquisition, K.K. (Kazunori Koga) and M.S. All authors have read and agreed to the published version of the manuscript.

**Funding:** This study was partly supported by JSPS KAKENHI Grant Number JP20H00142, JP20J13122 and JSPS Core-to-Core Program JPJSCCA2019002.

**Institutional Review Board Statement:** Not applicable.

**Informed Consent Statement:** Not applicable.

**Data Availability Statements:** Data is contained within the article.

**Acknowledgments:** This study was encouraged by Advanced Characterization Platform of the Nanotechnology Platform Japan sponsored by the Ministry of Education, Culture, Sports, Science and Technology (MEXT), Japan.

**Conflicts of Interest:** The authors declare no conflict of interest.

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
