# Peer review of "Time of Flight Size Control of Carbon Nanoparticles Using Ar+CH4 Multi-Hollow Discharge Plasma Chemical Vapor Deposition Method"

_processes, doi:10.3390/pr9010002_

Round 1
Reviewer 1 Report
Paper scope is the growth mechanism and size control of carbon nanoparticles generated in the gas flow in multi-hollow discharge plasmas.
Topic is interesting as nanotechnology is advancing in almost every science discipline and proposed paper gives some solution to acute problems of NPs synthesis.
Generally, the title of the paper is informative; quality of figures is fine, the length of the manuscript is suitable and the number of references is sufficient. Language is proper. Paper requires only minor “cosmetic” revisions as follows:
- Please add example of electrical characteristics of the plasma reactor with working gas.
- Please add description (arrows) to Fig. 1 depicting Si substrate and CNPs.
- Minor editorial mistakes for instance line 67- subscript in CH4
Author Response
We would like to thank the reviewer for the useful comments.
Please kindly find our response to the comments.
Comment 1: Please add example of electrical characteristics of the plasma reactor with working gas.
Response: As the reviewer suggested, we have added the sentences describing the discharge's electrical characteristics. The discharge voltage was 230 V, and self-bias voltage was 80 V. We have added this information on page 2 and line 75.
Comment 2. Please add description (arrows) to Fig. 1 depicting Si substrate and CNPs.
Response: As the reviewer suggested, we added the description of substrates (Si substrate or grid mesh for TEM) and CNPs in Figure 1 on page 2 and line 77.
Comment 3. Minor editorial mistakes for instance line 67- subscript in CH4
Response: We would like to thank the reviewer for pointing out our mistakes. We have corrected it on page 2 and line 68 in the revised manuscript.
Reviewer 2 Report
The manuscript presents study on effect of flow rates on Carbon nano particle size control. The work is very interesting and well written manuscript. Some English mistakes typical of non-English speaking authors were noticed.
It is recommended for acceptance after thorough check of English grammar.
Author Response
We would like to thank the reviewer for the useful comments.
Please kindly find our response to the comments.
Comment 1. It is recommended for acceptance after thorough check of English grammar.
Response: As the reviewer suggested, An English native speaker checked the revised manuscript.
Reviewer 3 Report
The work by Sung Hwa Hwang et al. deals with very interesting topic, especially that the technique can be upgraded to other materials or modification.
Generally manuscript is written well and most of the results and discussion is correct .
However, authors dont pay to much attention to character of prepared nanoparticles. Figure 2 is having 4 parts but it is not explained. From TEM is visible that particles has some core-shell particles. Explaining the character of the nanoparticles would be important. Examination using X-ray diffraction would explain real phase and structure of particles ,and more precise Raman will be complimentary.
There are several typos in manuscript such as indexes and units , I would suggest to use Pa instant of Torr.
Author Response
We would like to thank the reviewer for the useful comments.
Comment 1. Figure 2 is having 4 parts but it is not explained.
Response: As reviewer suggested, we added more explanation about Fig. 2 on page 2 and lines 87-98.
Comment 2. From TEM is visible that particles have some core-shell particles.
Response: We re-checked the TEM image and observed that most particles do not have the core-shell particles. As the reviewer mentioned, some particles have core-shell, but there are in minimal quantity. Currently, we have no explanation for the generation of such particles, but we will investigate these particles in the future. We are thankful to the reviewer for his/her comments.
Comment 3. Explaining the character of the nanoparticles would be important. Examination using X-ray diffraction would explain real phase and structure of particles, and more precise Raman will be complimentary.
Response: As the reviewer suggested, we add the X-ray diffraction to explain the structural character of the particles, and we observed amorphous carbon nanoparticles. We have added this information on page 4, line 122-126, page 5, lines 127- 130, and Figure 5.
Comment 4. There are several typos in the manuscript such as indexes and units, I would suggest to use Pa instant of Torr.
Response: We would like to thank the reviewer for pointing out our mistakes regarding indexes and units. We have corrected them in the whole manuscript and changed the pressure unit from Torr to Pa.